# CONTROLLING LANGUAGE OVER-OPTIMIZATION BY TARGETING REWARD DISTRIBUTION

## ABSTRACT

Reinforcement Learning (RL) has become a key optimization tool for fine-tuning and aligning Large Language Models (LLM) with human preferences. However, this approach relies on reward models susceptible to reward over-optimization, wherein language models learn to hack the reward function, resulting in unnatural generations. In this paper, we address this issue by aligning the reward distribution of sentences generated by the fine-tuned model with a predefined target reward distribution. It offers an a priori and parameter-free control over the distribution of rewards of the model, setting it apart from other regularization and post-processing techniques. Our experiments show that this RL approach alleviates several optimization challenges in LLM: it reduces the log-likelihood error accumulation when generating lengthy sequences, mitigates reward hacking when generating positive reviews on IMDB, and upholds length constraints while aligning summaries with human preferences on the TL;DR dataset. Our findings highlight that targeting reward distributions is a promising strategy to better control and enhance the reliability of RL-based fine-tuning.

## 1 INTRODUCTION

Reinforcement Learning (RL) is a key tool in modern Large Language Models (LLM) (Ramamurthy et al., 2022). Unlike supervised learning methods, RL enables optimizing non-differentiable objectives at the sequence-level, which are prevalent in the Natural Language Processing (NLP) (Ranzato et al., 2015; Paulus et al., 2017) and goal-oriented dialogue literatures (Wei et al., 2018; Strub et al., 2017). Recently, RL has successfully enhanced the alignment of LLMs with human preferences through reward models derived from Human Feedback (RLHF) (Christiano et al., 2017; Stiennon et al., 2020; Ouyang et al., 2022). This development has resulted in the creation of conversational assistants that are both more helpful and harmless (OpenAI, 2023; Bai et al., 2022).

However, fine-tuning LLMs with RL can be harmful if not carefully controlled. It may lead to a reduction in model diversity due to over-optimization toward the reward model (Gao et al., 2023), or to generation of unnatural language patterns to artificially inflate rewards (Paulus et al., 2017), or even alterations in the semantics and syntax of the initial language due to lack of grounding (Lewis et al., 2017). Until now, this phenomenon has been mitigated by incorporating a KL regularization term to anchor the fine-tuned model to its initialization. However, achieving the right calibration of the KL term — and consequently the final model — requires careful tuning of various hyperparameters (Stiennon et al., 2020). Ultimately, the incorporation of KL regularization can be viewed as an *a posteriori* selection process, as we only discern its effects after the training process.

In this work, we introduce an *a priori* methodology to mitigate reward over-optimization and better calibrate the RL-based fine-tuning of LLMs. Our approach focuses on aligning the reward of the generated sentences with a predefined target distribution. In essence, we optimize the model to conform to a *distribution* of rewards, rather than directly optimizing the reward itself, avoiding unintended collateral reward optimization. This introduces an *a priori* and parameter-free mechanism for governing the reward distribution of the fine-tuned model, setting our method apart from traditional regularization and post-processing techniques. Our approach hinges on a straightforward modification of the reward, which can be directly optimized in conjunction with conventional RL algorithms.

We validate our approach through three experiments, each highlighting an optimization challenge. We first motivate the importance of targeting a reward distribution to optimize sequence-level log-

likelihood. This is crucial as teacher-forcing pre-training can introduce errors during generation. RL can address this issue by correcting the log-likelihood, but it requires precise control to avoid excessively high values indicative of repetitive patterns (Holtzman et al., 2019). Our approach tackles this problem by aligning the fine-tuned policy with the target log-likelihood distribution of human sentences, ensuring log-likelihood stability during generation. In the second experiment, we apply our method to address reward over-optimization in a conventional reward model pre-trained on IMDB reviews. Remarkably, our method efficiently optimizes the reward without over-specialization, without the need for hyperparameter tuning. In contrast, achieving similar results by directly maximizing the reward requires an extensive cross-validation process for KL hyperparameters to match the final distribution. Lastly, we explore the benefits of targeting the reward distribution in a multi-reward setting, specifically in summarization, using a human preference model while incorporating length constraints into the reward function. Again, our parameter-free approach demonstrates its effectiveness for LLM fine-tuning, not requiring an extensive hyperparameter search.

## 2 RELATED SECTION

**Reinforcement learning for Natural Language Processing** Along with the rise of Large Language Models (LLM) (Brown et al., 2020; OpenAI, 2023; Anil et al., 2023; Touvron et al., 2023b), Reinforcement Learning (RL) has become a key optimization technique to fine-tune language models (LM) (Ramamurthy et al., 2022). Initially used to perform sequence-level optimization (Ranzato et al., 2015) or train non-differentiable objectives such as evaluation metrics (Wu et al., 2016; Wu & Hu, 2018) or signals from other networks (Kryściński et al., 2019; Scialom et al., 2020), RL has become standard to align LMs with human preferences through Reinforcement Learning with Human Feedbacks (RLHF) (Christiano et al., 2017). RLHF reward models (RM) are trained on pairwise comparisons of human sentences and then used as proxies to score LMs' generations. This approach has demonstrated its effectiveness across a range of tasks, including summarization (Stiennon et al., 2020), instruction following (Ouyang et al., 2022) or question answering (Nakano et al., 2021). Within the context of conversational assistants, RLHF is also used to control LM's generations on diverse criteria such as helpfulness or harmlessness (Bai et al., 2022; Glaese et al., 2022).

**The burden of training RL objectives in language** Language Reward models are imperfect proxies of the target task they try to reward. First, approximating such a reward function is inherently challenging and flawed due to the vast intricacies of language which is computationally untractable (Schatzmann et al., 2006). Therefore, reward models may only approximate the underlying language distribution, and may have blindspots, biases, etc. Second, it is challenging to capture a specific language behavior through metrics even when the task is clearly defined (Clark & Amodei, 2016). Therefore, maximizing this approximate reward signal with RL methods often leads to unintended consequences, empirically ranging from reward over-optimization (Gao et al., 2023; Amodei et al., 2016), which results in a loss of generality and the generation of language artifacts to artificially inflate reward scores (Paulus et al., 2017), to language drift in the multi-agent setting (Lazaridou et al., 2020; Lu et al., 2020), which results in alterations of language semantics and syntax due co-adapatation and lack of grounding. Overall, there is a long list of papers that observed diverse facets of this optimization pitfall (Everitt & Hutter, 2016; Zhuang & Hadfield-Menell, 2020; Skalse & Abate, 2022) The most standard approach to mitigate this issue is to tie the fine-tuned model to its origin through a KL pressure (Ziegler et al., 2019; Stiennon et al., 2020; Ouyang et al., 2022; Bai et al., 2022; Jaques et al., 2019). It ensures that the fine-tuned model does not deviate too far from its original pretrained policy. This regularization prevents the model from finding a solution that is highly rewarded by the RM while producing unnatural language.

In this paper, we aim to cover various applications of RL for LM training. We distinguish three classes of RL fine-tuning challenges from which we take one example from the literature: an RM used for sequence-level optimization, namely the LM itself as an evaluator of the sequence-level log-likelihood; an RM that is susceptible of reward hacking, namely a sentiment classifier trained on the IMDB dataset (Maas et al., 2011); the optimization of two reward models including an RLHF reward model, namely an RM designed to enhance summarization performance (Stiennon et al., 2020).

## 3 METHODOLOGY: TARGETING REWARD DISTRIBUTION

This section introduces the rationale behind and the implementation of the target reward approach.

### 3.1 KL-REGULARIZATION TOWARD LIMITING LLM DISTRIBUTION SHIFT

RL maximizes the expected reward of the policy $\pi$ given an environment, and its underlying reward function (Sutton & Barto, 2018). By parametrizing the policy with $\theta$, the problem can be written as:

$$\pi_{\theta^*} = \arg\max_{\theta} \mathbb{E}_{x \sim \mathcal{D}, y \sim \pi_\theta(.|x)} \left[ R(x, y) \right] \tag{1}$$

where $\pi_{\theta^*}$ is the optimal policy, $x \sim \mathcal{D}$ are the trajectories of states sampled from the initial state and $y$ are the actions picked by the policy at a given state.

When applied to LM, the reward function $R$ scores generated text on a predefined prompt or task. The reward typically encapsulates user preferences, as exemplified in RLHF. Yet, direct optimization of this reward model may lead to overoptimization, yielding to a poor finetuned model (Gao et al., 2023). So far, the prevailing method for preventing the LM policy from exploiting the reward model has been to introduce a KL regularization term to regulate the divergence between the fine-tuned policy and the initial one (Ziegler et al., 2019). By avoiding the distribution shift of the LLM policy, this term limits reward over-optimization. As multiple KL variants co-exist in the literature, we here focus on the empirical reward-based KL as in (Ziegler et al., 2019)

$$R_{reg}(x, y) = R(x, y) - \beta \log \left( \frac{\pi_\theta(y|x)}{\pi_{\theta_0}(y|x)} \right) \tag{2}$$

where $\pi_{\theta_0}$ is the initial policy and $\beta$ the parameter controlling the influence of the regularization.

KL-based regularization methods have several drawbacks: tuning the hyperparameter $\beta$ is task-specific and computationally expensive as it requires cross-validation, and there is no a-priori control over the final policy. In this paper, we propose a straightforward and intuitive approach. Instead of limiting the distribution shift when maximizing the reward, we directly target a desired expected reward distribution. By defining such a target, we provide an a priori control on the optimal solution and avoid hyperparameter tuning.

### 3.2 OUR APPROACH

We here define a protocol toward aligning a fine-tuned model with a target reward distribution, which can be summarized as follows:

1. Collect a distribution of rewards $R(x, y)$ from input/output pairs $(x, y)$
2. Define a reward function $R_{target}$ enforcing the alignment with the target distribution.
3. Optimize the LLM to maximize the reward $R_{target}$ using any RL algorithm.

**Step 1: Defining the target reward distribution:** The distribution of natural language is known to encompass a wide range of patterns (Zipf, 1949). By extension, reward distributions should also exhibit diverse patterns, composed of a mix of very good, excellent, and exceptional scores. Targeting a Dirac distribution that exclusively assigns exceptional scores (as in RLHF) is unrealistic without resorting to some exploitation. In other words, striving for perfectionism can prove counterproductive. With this in mind, we compile a set of prompt-continuation pairs $(x, y)$ from human demonstrations and evaluate them with the reward model. This establishes the natural target reward $R(x, y)$ for prompt $x$. This process creates a prompt-reward pair dataset $\mathcal{D} := (x, t(x))$, which encodes the natural distribution of target rewards. In this paper, we apply simple approaches to build a suitable prompt-continuation dataset based on publicly available data. In other RLHF contexts, one may re-label a user-preference dataset, e.g. retaining only the best components in pairwise comparisons.

**Step 2: Defining the reward function:** Given the prompt-reward dataset $\mathcal{D} := (x, R(x, y))$, the LLM generates a continuation $\hat{y}$ of the prompt $x$ that is scored $R(x, \hat{y})$. Instead of maximizing $R(x, \hat{y})$ directly, we modify the problem into a non-differentiable regression with a negative $L_2$ objective:

$$R_{target}(x, \hat{y}) = -||R(x, \hat{y}) - R(x, y)||_2^2 \tag{3}$$

Consequently, the LLM is incentivized to get a reward close to its target. This differs from the standard approaches in the literature where the objective is to directly maximise $R(x, \hat{y})$ (or most commonly its regularized counterpart, $R_{reg}(x, \hat{y})$).

**Step 3: LLM Optimization:** As the reward objective is non-differentiable, it can be optimized by using policy gradient methods. In this paper, we use PPO gradient update (Schulman et al., 2017).

## 4 EXPERIMENTAL SETTING

In this paper, we explore three distinct LLM finetuning settings in which a direct maximization of the reward model turns out to be challenging. In each case, we later show that targetting a reward distribution enables better control of the optimization process while successfully achieving the task.

### 4.1 USE CASE 1: CALIBRATING SEQUENCE-LEVEL LOG-LIKELIHOOD

**Motivation:** Although LLMs are optimized to maximize their per-token log-likelihood, they accumulate errors when generating long sequences at inference time. This phenomenon is often described to be an artifact of the exposure bias (Ranzato et al., 2015). However, attempts to suppress this training artifact by maximizing the average sequence-level log-likelihood, e.g., with beam-search, resulted in degenerate solutions (Holtzman et al., 2019). As a proof of concept to our approach, we here explore whether it is possible to remove this error accumulation by targeting sequence-level log-likelihood distribution rather than directly maximizing the sequence-level log-likelihood as previously attempted.

**Setup:** We use the Wikipedia dataset (Wikimedia, 2023) where the prompt-continuation $(x, y)$ pairs are split sentences of respective lengths of $64$ and maximum $320$ tokens. The LM policy is a LlaMa2-7B (Touvron et al., 2023b). The reward model is another frozen LlaMa2-7B, which computes the sequence-level log-likelihood of the ground-truth continuation $y$ and the policy continuation $\hat{y}$, outputting both $t(x) = R(x, y)$ and $R(x, \hat{y})$. Notably, the policy may generate up to $320$ tokens during training but is evaluated with generations of up to $1000$ at evaluation time.

**Baseline:** We compare our method with two classes of baseline models. First, the initial LM with different decoding heuristics, namely greedy decoding, nucleus sampling, and temperature sampling. Second, models fine-tuned with standard RL approaches: i.e., maximising $R$ and $R_{reg}$. We refer to these approaches as $R$ and $R_{reg}$ respectively.

### 4.2 USE CASE 2: MITIGATING OVER-SPECIALIZATION WITH A CLASSIFIER REWARD MODEL

**Motivation:** Aligning LLM with reward models may be performed with various pretrained reward models ranging from a classic language classifiers (Ramamurthy et al., 2022), estimating a ELO score over human sentences (RLHF) (Ouyang et al., 2022), or even sampling a finetuned LLM (RLAIF) (Lee et al., 2023). Yet, all those models are subject to reward over-specialisation (Ziegler et al., 2019). We here assess how targeting a reward model distribution may mitigate over-specialisation when aligning an LLM with a generic language classifier (Lee et al., 2023).

**Setup:** As in (Ramamurthy et al., 2022), we use a sentiment analysis model trained on the IMDB dataset (Maas et al., 2011). We use an LlaMa2-7B as the LM policy (Touvron et al., 2023b) and a DistilBERT (Sanh et al., 2019) from HuggingFace as the reward model[1]. We take the output logit associated with the positive class as reward $R(x, y)$. Prompts are based on the first 10 tokens of the IMDB dataset's positive reviews for training and validation and are kept short to reduce positive signals before generation. The policy may generate up to 160 tokens.

**Baseline:** We compare our method with standard fine-tuning procedures, namely SFT, $R$ and $R_{reg}$.

### 4.3 USE CASE 3: CALIBRATING MULTI-OBJECTIVE RLHF SYSTEMS

**Motivation:** Many RLHF settings combine multiple reward signals, such as toxicity metrics (Glaese et al., 2022) and helpfulness vs. harmfulness (Bai et al., 2022). When maximizing the reward, these systems can produce two calibration issues: (1) the policy may over-specialize on one of the reward

---

[1]https://huggingface.co/lvwerra/distilbert-imdb

models, and (2) the policy may favor some reward models over others. Therefore, exploring the Pareto frontiers over multiple rewards has been challenging (Rame et al., 2023). In this setting, we show that controlling and targeting the reward distribution enables us to anticipate how the policy will balance rewards and where the fine-tuned LLM may land on the Pareto frontier beforehand.

**Setup:** We study the multi-reward finetuning problem on a summarization task under constraint on the TL;DR Reddit dataset (Völske et al., 2017). In particular, we combine a human-preference score with a length penalty as summarization reward models may not always capture concision (Lee et al., 2023). To do so, we first estimate a user-preference summarization score $R_{pref}(x, y)$ by using the DeBerta reward model (He et al., 2020) from OpenAssistant (Köpf et al., 2023). We then compute the number of generated tokens to get $R_L(x, y)$ and define the final reward as $R(x, y) = \alpha_{pref}R_{pref}(x, y) - \alpha_L R_L(x, y)$. As our initial language policy, we use the Alpaca model (Taori et al., 2023), which is a finetune version of LLaMa-7B (Touvron et al., 2023a) that can perform summarization. Taking inspiration from (Lee et al., 2023), we create a training prompt-continuation dataset by keeping the chosen summary with the annotator confidence above 5 for a total of 22k pairs. By doing so, we estimate a high-quality target reward distribution that jointly encapsulates the high user-reward preference scores and the expected length-penalty of well-written summaries.

**Baseline:** We compare our method with the standard fine-tuning protocols, namely SFT, $R$ and $R_{reg}$.

## 4.4 TRAINING AND EVALUATION

During finetuning, we use Low Rank Adaptation (LoRA) (Hu et al., 2022) with PEFT (Mangrulkar et al., 2022), and Adam optimizer (Kingma & Ba, 2014). In each case, we report the best model in terms of alignment after performing a grid search over the learning rate, batch-size, LoRA parameters, and the KL coefficient $\beta$ (RL only). Hyperparameters are in Appendix A and the code is available at **HIDDEN**. For each use-case, we evaluate the finetune model over four criteria as detailed below:

**Alignment:** We measure whether the reward distribution of the fine-tuned LLM is aligned with the human reward distribution over the validation set. Formally, given the normalized distribution $d_{\pi_\theta}$ of rewards obtained when generating the continuation of prompts with stochastic sampling, and the normalized distribution of rewards of the ground truth human continuations of the validation set $d_H$, we define the alignment score $\mathcal{A}$ as the KL distance between the two distribution, i.e., $\mathcal{A} = D_{KL}(d_H|d_\pi)$. The lower $\mathcal{A}$, the more $d_\pi$ and $d_H$ are aligned, with an optimal score of 0.

**Task Success:** We evaluate task success with an AI feedback process conducted by chat-LlaMa2-13B (Touvron et al., 2023b). For every task, we prompt chat-LlaMa2-13B to classify whether the given output accomplishes the task and report the average score over the validation set. Prompt templates are provided in Appendix B.

**Naturalness:** We evaluate how human-like (or natural) is a generated sentence with an AI feedback process conducted by chat-LlaMa2-13B (Touvron et al., 2023b), and report the average score over the validation set. Prompt templates are provided in Appendix B.

**Diversity:** To evaluate generation diversity, we calculate the type/token ratio, which is the ratio between the number of unique words and the total number of words in the generated text as in (Xu et al., 2022; Lin et al., 2021). Specifically, we refer to the type/token ratio at the sentence level as S-Diversity and the ratio at the corpus level as C-Diversity. These metrics allow for detecting unnatural and repetitive patterns in the generated text.

## 5 RESULTS

In this Section, we derive the results of the three use cases: sequence-level log-likelihood calibration problem, classic RL fine-tuning setting, and multi-reward optimization in an RLHF context.

## 5.1 USE CASE 1: CALIBRATING SEQUENCE-LEVEL LOG-LIKELIHOOD

**Log-likelihood vanishes along the generation** In Figure 1a, we observe that the average per-token log-likelihood of sentences generated by a LLM (with $\tau = 1$) diminishes with longer sentences (green line), highlighting the persistence of the *exposure bias* (Ranzato et al., 2015) in LLMs (-17% after 700 tokens as reported in Table 1). On the contrary, human-generated sentences maintain

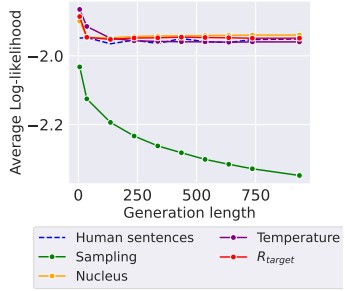

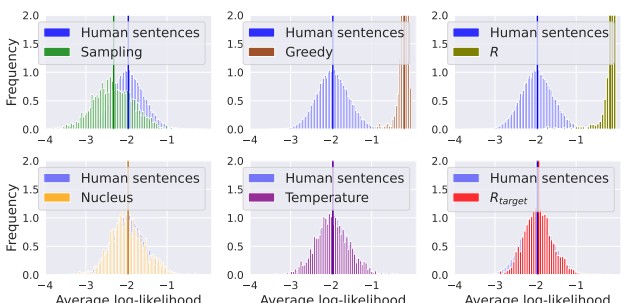

(a) Average log-likelihood as a function of the generation length

(b) Distribution of the average log-likelihood of human sentences over the different baselines (generations of 700 tokens)

| Method | Alignment ↓ | Log-likelihood ↑ | Naturalness ↑ | S-Diversity ↑ | C-Diversity ↑ |
|---|---|---|---|---|---|
| Human sentences | - | -1.95 | 0.73 | 0.61 | 0.05 |
| *Sampling strategy* | | | | | |
| Temperature ($\tau = 1$) | 0.33 | -2.28 | 0.20 | **0.56** | **0.10** |
| Temperature ($\tau = 0.96$) | **0.04** | **-1.96** | 0.32 | 0.53 | 0.09 |
| Temperature ($\tau = 0$) | 0.98 | -0.26 | 0.03 | 0.13 | 0.03 |
| Nucleus ($p = 0.97$) | 0.06 | **-1.96** | **0.33** | 0.52 | 0.08 |
| *Training strategy* | | | | | |
| $R$ | 1.07 | -0.19 | 0.01 | 0.07 | 0.03 |
| $R_{target}$ | **0.04** | **-1.94** | 0.32 | 0.52 | 0.08 |

Table 1: Scores of the Use Case 1. Best scores among models are in bold, second best underlined, bad outliers in gray. Alignment and log-likelihood are reported for generations of up to 700 tokens.

a consistent per-token log-likelihood (dashed blue line) even for long sentences. This disparity is also depicted in the gap between the sentence-level log-likelihood distributions of human and LLM-generated sentences (see Figure 1b). In sum, our findings corroborate earlier findings on smaller models (Holtzman et al., 2019).

**From maximizing to calibrating sequence-level log-likelihood** To mitigate the decrease of log-likelihood with sequence length, we maximize the estimated sequence-level log-likelihood using a frozen copy of the LM as RM. In Figure 1b and Table 1, we note that finetuning by maximising $R$ successfully reach a very high likelihood of ($-0.19$ in average), and exceeding human likelihood ($-1.96$ in average). Yet, the resulting policy generates unnatural and repetitive sentences, as highlighted by the significant decrease of both *S-Diversity* and *C-Diversity* (Table 1) and observed in samples in Appendix C.1. Essentially, the fine-tuned policy learns to hack the RM, a phenomenon also evident in Figure 1b through a shift in the reward distribution between LM-generated sentences and human-generated sentences. However, when targeting the log-likelihood distribution with $R_{target}$, we observe that the average log-likelihood matches the human one (1% difference). More impressively, the LM fine-tuned with $R_{target}$ has a reward distribution that is fully aligned with the human reward distribution on the evaluation set in Figure 1b with an alignment score of $0.04$. However, we do not manage to retrieve the same diversity, mining that this fine-tuning may fix the log-likelihood gap by tightening the policy distribution. Finally, Figure 1a illustrates that using $R_{target}$ enables to resolve the dismishing log-likelihood over long sequences, therefore successfully calibrating the LLM.

**Comparison with baseline decoding heuristics** As shown in Figure 1a, tuned decoding methods, namely temperature sampling and nucleus sampling, can effectively calibrate sentence-level log-likelihood and maintain it close to the level of human-generated sentences. This is consistent with previous work (Holtzman et al., 2019). Figure 1b illustrates that these heuristics effectively realign the human and LM log-likelihood distributions. Notably, greedy decoding (temperature=0) yields the same issue as maximizing $R$: both methods show poor results in terms of naturalness and diversity. However, while temperature and nucleus sampling can successfully realign the distributions, they are model-dependent and require an extensive search over the sampling hyperparameters. In contrast, targeting the log-likelihood distribution is parameter-free and model-independent.

## 5.2 USE CASE 2: MITIGATING OVER-SPECIALIZATION WITH A CLASSIFIER REWARD MODEL

Figure 2: Comparison between the reward distribution of human sentences and LM generations for the different methods. Vertical lines mark the mean of the distribution.

| Method | Alignment $\downarrow$ | Av. $R \uparrow$ | Success $\uparrow$ | Naturalness $\uparrow$ | S-divesity $\uparrow$ | C-Diversity $\uparrow$ |
|---|---|---|---|---|---|---|
| Human sentences | - | 2.01 | 0.94 | 0.73 | 0.59 | 0.02 |
| *Without RL* | | | | | | |
| No fine-tuning | 0.33 | 1.10 | 0.73 | 0.69 | **0.80** | **0.07** |
| SFT | 0.10 | 1.58 | 0.77 | 0.60 | 0.60 | 0.04 |
| *With RL* | | | | | | |
| $R$ | 1.38 | **2.96** | **0.94** | 0.09 | 0.43 | 0.01 |
| $R_{reg}(\beta = 0.1)$ | 0.51 | 2.52 | **0.94** | **0.74** | 0.68 | 0.05 |
| $R_{reg}(\beta = 0.3)$ | 0.04 | 2.01 | 0.89 | 0.66 | 0.69 | **0.07** |
| $R_{target}$ | **0.04** | 1.97 | 0.93 | 0.68 | 0.63 | 0.06 |

Table 2: Use Case 2's score. Best scores are in bold, second best underlined, bad outliers in gay.

**Reward over-optimization and standard KL regularization** Figure 2 shows that the policy fine-tuned to maximise $R$ successfully increases the positive score of the sentiment RM. However, the resulting reward distribution notably diverges from human sentences' rewards distribution (in blue), indicating a *reward over-optimization* regime. Upon analyzing specific generated examples (See Appendix C.2), we observe that the fine-tuned language model tends to produce unnatural repetitions of positive adjectives, which is also reflected by low naturalness and diversity scores (Table 2). Introducing a KL penalty with a tunable parameter $\beta$ is effective in mitigating this effect (see Figure 2 with $\beta = 0.1$ and $\beta = 0.3$). With careful tuning of $\beta$, we show that KL regularization aligns the reward distribution of the fine-tuned policy with that of human positive reviews. It reaches a very low alignment score of $\mathcal{A} = 0.04$. For completeness, we also report results with SFT, which successfully generates positive reviews with high success scores. This is also reflected by a significant alignment ($\mathcal{A} = 0.10$) of its reward distribution with human-generated sentences' reward.

**Countering reward over-optimization with target rewards** Alternatively to KL regularization, using $R_{target}$ offers a direct means to calibrate the fine-tuned policy. Figure 2 shows that $R_{target}$ successfully guides the policy toward a model with a reward distribution aligned with the human distribution ($\mathcal{A} = 0.04$). It produces high-quality reviews when looking at generated examples (See Appendix C.2), avoiding reward over-optimization. This observation is further confirmed in Table 2, as $R_{target}$ has high naturalness, S-Diversity, and C-Diversity scores, and matches highly-tuned $R_{reg}$ scores. More importantly, $R_{target}$ has the same success ratio as $R_{reg}$, without requiring KL parameter search nor the training artifacts.

In short, maximizing $R_{target}$ efficiently mitigates reward over-optimization and preserves performance without hyperparameter tuning, which is highly valuable when scaling up to large models.

## 5.3 USE CASE 3: CALIBRATING MULTI-OBJECTIVE RLHF SETTING

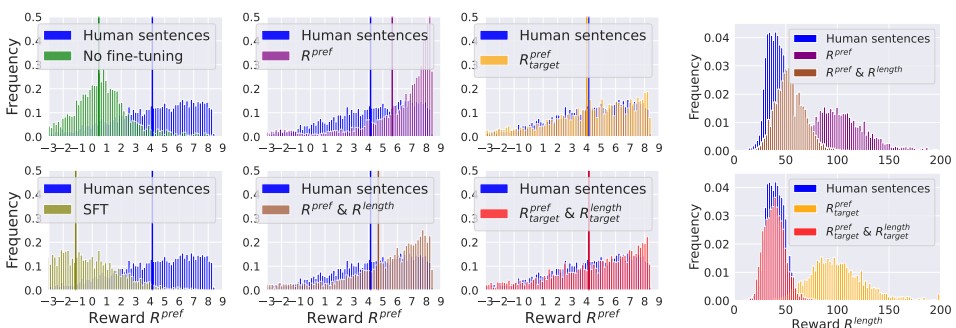

Figure 3: (left) Distribution of the rewards $R^{pref}$. (right) $R^{length}$ per fine-tuning method.

| Method | Alig. ↓ | $R^{pref}$ ↑ | $R^{length}$ ↑ | Success ↑ | Naturalness ↑ |
|---|---|---|---|---|---|
| Human sentences | - | 4.14 | 40.23 | 0.90 | 0.79 |
| No fine-tuning | 0.86 | 0.45 | 133.6 | 0.61 | 0.83 |
| SFT | 0.70 | -1.13 | 60.2 | 0.76 | 0.84 |
| *Single reward setting* | | | | | |
| $R^{pref}$ | 0.45 | 5.64 | 102.5 | 0.73 | 0.94 |
| $R^{pref}_{reg} (\beta = 0.3)$ | 0.43 | 1.91 | 101.2 | 0.71 | 0.81 |
| $R^{pref}_{target}$ | **0.11** | 4.01 | 107.1 | 0.74 | 0.89 |
| *Two-reward setting* | | | | | |
| $R^{pref}$ & $R^{length}$ | 0.18 | 4.68 | 54.5 | 0.70 | **0.97** |
| $R^{pref}_{target}$ & $R^{length}_{target}$ | **0.12** | **4.15** | **39.40** | **0.83** | **0.98** |

Figure 4: Use Case 3's score. Best scores are in bold, second best underlined, bad outliers in gray.

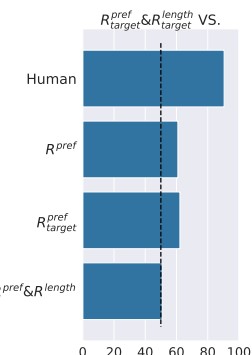

Figure 5: Win Rates for $R^{pref}_{target} / R^{length}_{target}$

Finally, we test our method in a more realistic scenario: summarization. We investigate both cases when we only optimize a reward model $R^{pref}$ trained with human preferences and when we optimize both $R^{pref}$ and an RM equal to the length of the generation $R^{length}$. This example raises two challenges: preventing reward over-optimization and co-optimizing multiple rewards.

**Optimizing only** $R^{pref}$ Figure 6 shows the distribution of $R^{pref}$ for different fine-tuned LM. We first note that the distribution of the initial model is distributed around 0, meaning that fine-tuning is truly required. Second, we observe that contrary to use case 2, applying SFT does not correct the distribution of rewards. Third, $R^{pref}$, expectedly, leads to a significant increase of the reward (5.64). We should note though that the resulting distribution with $R^{pref}$ is less concentrated around the argmax compared to our prior use cases. This may be explained by the RM which was trained on several datasets and hence harder to "hack", or by the use of LoRA fine-tuning which may regularize the training and prevent overfitting. This absence of over-optimization is reflected in concrete examples and metrics: the naturalness score stays at a high value (0.94 compared to 0.79 for human sentences). Still, the reward distribution obtained with $R^{pref}$ is hard to control and does not align with our ground truth distribution ($\mathcal{A} = 0.45$). Furthermore, KL regularization is harder to set than in the previous case as it is either ineffective or excessive. Finally, once again, when training with $R^{pref}_{target}$, the fine-tuned policy distribution aligns with the ground truth distribution which is reflected by a gain of alignment ($\mathcal{A} = 0.11$ compared to $\mathcal{A} = 0.45$ with $R$ (see Table 4).

Even if both $R^{pref}$ and our approach correctly optimize the reward, we note in Figure 6 that fine-tuned models optimizing the preference reward only generate summaries significantly longer than human

summaries. This result is in line with previous works using this dataset (Stiennon et al., 2020; Lee et al., 2023): The RM may not capture the inherent constraints of producing short generations when summarizing which motivated the introduction of an additional length constraint.

**Multi-reward calibration** When introducing the length constraint $R^{length}$, we turn to a 2-reward optimization problem. When training with $R^{pref}$ and $R^{length}$, carefully tuning the balance between the two rewards is required to prevent the model from focusing on a single reward or collapsing. Adjusting the weights of the two rewards leads to a compute-expensive and controless process that does not even guarantee a correct alignment. Yet, we performed such tuning and came up with a model that produces generations with quite similar lengths as the target distribution while keeping a high reward. Using two target rewards $R^{pref}_{target}$ and $R^{length}_{target}$ instead is well suited for this kind of problem. Tuning is limited to a normalization of the two rewards and the targets naturally guide the fine-tuned policy toward a well-calibrated reward distributions. Table 4 and Figure 3 show that the fine-tuned policy with the target rewards achieves the best alignment, closely matching the human mean reward and obtaining both high success and naturalness scores. We eventually compute the Win Rate (WR) (Lee et al., 2023) between models. LlaMa2-13B (Touvron et al., 2023b) is prompted (see the template in Appendix B) to select its preference between two generated sequences for a given input. The WR is then averaged as the percentage of chosen generations of a model with respect to another. Figure 5 displays the WR of the policy fine-tuned with $R^{pref}_{target}$ and $R^{length}_{target}$ against other models. Results show that this policy is significantly more preferred than ground truth human summaries ($\sim 90\%$) and than models fine-tuned on $R^{pref}$ or $R^{pref}_{target}$ only ($\sim 60\%$).

## 6 DISCUSSION

**Target rewards VS. KL regularization** In our experiments, we show that carefully tuning the KL regularization may achieve good alignment with the human reward distribution. However, directly optimizing $R_{target}$ brings more control; in this case, tuning is then limited to pure optimization and is often costless. In the context of LLMs and RLHF, fine-tuning is very costly. Therefore, being able to have control over the fine-tuning process is important. This makes our approach promising compared to standard KL regularization. Additionally, it is worth mentioning that $R_{target}$ and KL are not mutually exclusive and could be combined during training.

**Building target distributions** In this work, we focused on building data-driven target distributions. Note that we explored simple examples. In realistic scenarios, the target distribution should rely on high-quality data (e.g. human selected sentences that are both harmless and helpful in a harmlessness/helpfulness 2-reward optimization (Bai et al., 2022)). Note that in principle our method could be applied to any hand-crafted target distribution. For example this would allow incorporating specific constraints on the reward distribution (via eg. truncation). While preliminary experiments (outlined in Appendix D) show that optimising discontinuous distribution poses new challenges, we believe that this is a promising direction for future research.

## 7 CONCLUSION

In this paper, we propose a new method for better controlling RL fine-tuning in LLMs and avoid reward over-optimization. Our method aligns the distribution of rewards of the fine-tuned policy with a target distribution. This approach differs from standard regularization methods as it is parameter-free and does not require model selection, which is crucial when scaling up to large LLM where grid search is computationally impractical. We also demonstrate that targetting a distribution also greatly simplifies optimizing a mixture over rewards. This practical use-case is even more challenging when using specialized reward models that do not correlate with each others (Touvron et al., 2023b) leading to several Pareto equilibria (Rame et al., 2023).

Ultimately, directly maximizing reward models is an unrealistic endeavor due to their inherent imperfections. Our method proposes instead to target a specific given distribution, thus avoiding the pitfalls of over-optimization.

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

## A TRAINING HYPERPARAMETERS

In this Appendix, we report the technical details for all experiments and in particular the values of our hyperparameters.

### A.1 USE CASE 1

Parameters used for the Use Case 1 are gathered in Table 3.

| Experiment | $R(x,y)$ | $R_{reg}(x,y)$ | $R_{target}(x,y)$ |
|---|---|---|---|
| *Models* | | | |
| Policy | | LlaMa7B | |
| Reward model | | LlaMa7B | |
| *Optimizer* | | | |
| Type | Adam | Adam | Adam |
| learning rate | $5e-5$ | $5e-5$ | $5e-5$ |
| batch size | 25 | 25 | 25 |
| Accumulation steps | 20 | 20 | 20 |
| *LoRA* | | | |
| rank | 32 | 32 | 32 |
| $\alpha$ | 64 | 64 | 64 |
| dropout | 0.01 | 0.01 | 0.01 |
| bias | None | None | None |
| *PPO* | | | |
| $\epsilon$ | 0.3 | 0.3 | 0.3 |
| baseline | True | True | True |
| $\beta$ | 0.3 | 0 | 0 |

Table 3: Hyper-parameters for Use Case 1

#### A.1.1 USE CASE 2

Parameters used for the Use Case 1 are gathered in Table 4.

#### A.1.2 USE CASE 3

Parameters used for the Use Case 3 are gathered in Table 5.

### A.2 ADDITIONAL DETAILS

- Note that the baseline used is: $R \leftarrow \frac{R-\sigma}{\eta}$ where $\sigma$ is the mean of the batch and $\eta$ is its standard deviation.
- The reward model used for Use Case 3 is available here: https://huggingface.co/OpenAssistant/reward-model-deberta-v3-large-v2

| Experiment | $R(x,y)$ | $R_{reg}(x,y)$ | $R_{target}(x,y)$ |
|---|---|---|---|
| *Models* | | | |
| Policy | | LlaMa7B | |
| Reward model | | DistillBERT | |
| *Optimizer* | | | |
| Type | Adam | Adam | Adam |
| learning rate | $5e-5$ | $5e-5$ | $5e-5$ |
| batch size | 25 | 25 | 25 |
| Accumulation steps | 20 | 20 | 20 |
| *LoRA* | | | |
| rank | 32 | 32 | 32 |
| $\alpha$ | 64 | 64 | 64 |
| dropout | 0.01 | 0.01 | 0.01 |
| bias | None | None | None |
| *PPO* | | | |
| $\epsilon$ | 0.3 | 0.3 | 0.3 |
| baseline | True | True | True |
| $\beta$ | 0 | 0.1→0.3 | 0 |
| $\alpha_{pref}$ | **0.11** | 0.74 | 0.89 |
| $\alpha_{length}$ | **0.11** | 0.74 | 0.89 |

Table 4: Hyper-parameters for Use Case 2

| Experiment | $R(x,y)$ | $R_{reg}(x,y)$ | $R_{target}(x,y)$ |
|---|---|---|---|
| *Models* | | | |
| Policy | | Alpaca 7B | |
| Reward model | | DeBerta (OpenAssistant) | |
| *Optimizer* | | | |
| Type | Adam | Adam | Adam |
| learning rate | $5e-5$ | $5e-5$ | $5e-5$ |
| batch size | 8 | 8 | 8 |
| Accumulation steps | 50 | 50 | 50 |
| *LoRA* | | | |
| rank | 32 | 32 | 32 |
| $\alpha$ | 64 | 64 | 64 |
| dropout | 0 | 0 | 0 |
| bias | None | None | None |
| *PPO* | | | |
| $\epsilon$ | 0.3 | 0.3 | 0.3 |
| baseline | True | True | True |
| $\beta$ | 0 | 0→0.3 | 0 |
| $\alpha_{pref}$ | 1 | - | 1 |
| $\alpha_{length}$ | 0.005 | - | 0.01 |

Table 5: Hyper-parameters for Use Case 3

## B  PROMPTS FOR EVALUATION

In this Section, we report the different templates of prompts used for our evaluations and discuss the protocol.

### B.1  TEMPLATES OF PROMPTS

**Task success**

- **Use Case 2**

> This first IMDB review has been written by a human:
> –GROUND TRUTH SUMMARY–
> Here is a second movie review:
> –GENERATION–
> Do you think it has also been written by a human ? Respond only Yes or No.

- **Use Case 3**

> A good summary provides all the information about a text while being concise. Here
> is a text to summarize:
> –TEXT TO SUMMARIZE–
> A good summary of it:
> –GROUND TRUTH SUMMARY–
> Here is a second summary:
> –GENERATED SUMMARY–
> Is the second summary a good summary ? Respond only Yes or No.

**Naturalness**

- **Use Case 1**

> This first Wikipedia article has been written by a human:
> –GROUND TRUTH EXAMPLE–
> Here is a second article:
> –GENERATION–
> Do you think it has also been written by a human ? Respond only Yes or No.

- **Use Case 2**

> This first IMDB review has been written by a human:
> –GROUND TRUTH EXAMPLE–
> Here is a second movie review:
> –GENERATION–
> Do you think it has also been written by a human ? Respond only Yes or No.

- **Use Case 3:**

> This first Reddit post has been written by a human:
> –GROUND TRUTH EXAMPLE–
> Here is a second post:
> –GENERATION–
> Do you think it has also been written by a human ? Respond only Yes or No.

**Win Rate**

- **Use Case 3**

> Choose the best summary. Respond only Summary 1 or Summary 2. Text:
> –TEXT–
> Summary 1:
> –GENERATION 1–
> Summary 2: –GENERATION 2–

B.2 REMARKS

To build our evaluation prompts, we did some prompt engineering. We first tried a 0-shot approach but on every Use case, ground truth human sentences obtained very low scores. In particular, we noted that prompting 0-shot on the TL;DR Dataset was very difficult because the model was expressing a negative opinion because of the ton of the data (even thou we were prompting the model about the quality of a summary).

We therefore adapted the protocol and used a 1-shot approach. Results were way more in line with our observations and the comments added by the model after the Yes/No answers were coherent. Still, we know that our metrics ara highly biased by the prompts. We estimate that it is mainly useful to detect extreme cases, such as degenerated policies (eg. the model trained with $R$ in Use Case 2 has a very low Naturalness score).

Moreover, when building the Win Rate metric for comparing summaries, we took care to randomly alternate the position of each model in the prompt order (Summary 1 or Summary 2 in the template) to avoid any bias linked to the order.

## C GENERATION EXAMPLES

Below, you'll find a collection of generated examples corresponding to each technique and use case. Please be aware that some of the generated content may be truncated because of the maximum length limit applied during the inference process.

### C.1 USE CASE 1

Prompt: **The Alexander Briggs House is a historic house located at 210 Jackson St. in Charleston, Illinois. Prominent local stonemason Alexander Briggs built the stone house in 1894; it is the only stone house remaining in Coles County. The house's design**

- Temperature ($\tau = 0$): **The Alexander Briggs House is a historic house located at 210 Jackson St. in Charleston, Illinois. Prominent local stonemason Alexander Briggs built the stone house in 1894; it is the only stone house remaining in Coles County. The house's design** is a combination of the Queen Anne and Richardsonian Romanesque styles. The house was listed on the National Register of Historic Places in 1982.

- Temperature ($\tau = 0.96$): **The Alexander Briggs House is a historic house located at 210 Jackson St. in Charleston, Illinois. Prominent local stonemason Alexander Briggs built the stone house in 1894; it is the only stone house remaining in Coles County. The house's design** is based on a 17th-century farmhouse in Llansteffan, Wales. Briggs wanted his high-end house to look like "an old Welsh home",[2] and it was both the home of Briggs and his wife Alma until 1914,[3] and the Charleston Public Library from 1940 to 1994. The house was added to the National Register of Historic Places on July 21, 1982. Stone house with hipped roof 139.37 acres (0.5653 km2)

- Temperature ($\tau = 1$): **The Alexander Briggs House is a historic house located at 210 Jackson St. in Charleston, Illinois. Prominent local stonemason Alexander Briggs built the stone house in 1894; it is the only stone house remaining in Coles County. The house's design** in 1894; it is the only stone house remaining in Coles County. The house's design was inspired by a book, Handbook of Architecture and Building, published in 1887 which presented 45 styles of American architecture. The style of the Alexander Briggs House is Neo-Georgian, resembling Classical American Georgian architecture. By the time Alexander Briggs built the house, his account book showed an impressive 24 grand buildings constructed upon request. Coles County's stone houses are representative of the stone craftsman movement that swept the United States during the late 19th century. Construction of the stone buildings occurred due to the abundance of stone quarries in the region.

- Nucleus ($\tau = 0.97$): **The Alexander Briggs House is a historic house located at 210 Jackson St. in Charleston, Illinois. Prominent local stonemason Alexander Briggs built the stone house in 1894; it is the only stone house remaining in Coles County. The**

**house's design** includes a hip roof, circular driveway, arched doorways, and windows. The house has been restored to its original condition and is currently used as a bed and breakfast. This page contains text from Wikipedia, the Free Encyclopedia - https://wn.com/Alexander-Briggs-House Latest News for: alexander briggs house Alexander Mantey, 16, was shot by police in the wake of ...... Federal Report: Central Dauphin West Safety Enforcement Slowed After Fatal Officer-Involved Shooting

- $R$: **The Alexander Briggs House is a historic house located at 210 Jackson St. in Charleston, Illinois. Prominent local stonemason Alexander Briggs built the stone house in 1894; it is the only stone house remaining in Coles County. The house's design** remaining in Coles County. The house's design was inspired by the Alexander Briggs House. The Alexander Briggs House is a historic house located at 210 Jackson St. in Charleston, Illinois. Prominent local stonemason Alexander Briggs built the stone house in 1894; it is the only stone house remaining in Coles County. The house's design was inspired by the Alexander Briggs House. The Alexander Briggs House is a historic house located at 210 Jackson St. in Charleston, Illinois. Prominent local stonemason Alexander Briggs built the stone house in 1894; it is the only stone house remaining in Coles County. The house's design was inspired by the Alexander Briggs House. The Alexander Briggs House is a historic house located at 210 Jackson St. in Charleston, Illinois. Prominent local stonemason Alexander Briggs built the stone house in 1894; it is the only stone house remaining in Coles County. The house's design was inspired by the Alexander Briggs House.

- $R_{target}$: **The Alexander Briggs House is a historic house located at 210 Jackson St. in Charleston, Illinois. Prominent local stonemason Alexander Briggs built the stone house in 1894; it is the only stone house remaining in Coles County. The house's design** features an open archway entry topped by a second-story round head window. The rear of the house is a bungalow addition from about 1915. Briggs was a skilled worker, specified in the building of both houses and essentially all common variable finished stone structures in the community. He also added touches of Gothic Revival architecture to the house. The house is a member of the Coles County Historic Register; Briggs is also a member of the Coles County Historic Register. The Alexander Briggs House has been home to intra-national amateur fencing championship matches.[2]

Prompt: **Pemberton and Shearsmith are fans of classic horror films, and the episode displays inspiration from the likes of Alfred Hitchcock's filmography and the Hammer Horror films. Pemberton called the episode the "most genre" of the series, while Shearsmith described it as**

- Temperature ($\tau = 0$): **Pemberton and Shearsmith are fans of classic horror films, and the episode displays inspiration from the likes of Alfred Hitchcock's filmography and the Hammer Horror films. Pemberton called the episode the "most genre" of the series, while Shearsmith described it as** "a bit of a horror film". The episode was written by Pemberton and Shearsmith, and was directed by Pemberton. The episode was first broadcast on 25 October 2012 on BBC Two. It received a mixed reception from critics, with some praising the episode's dark humour and the performances of the cast, while others criticised the episode's lack of originality and the episode's ending. The episode begins with a man (Jonathan Aris) and a woman (Sarah Hadland) driving through the countryside. The man is driving, and the woman is in the passenger seat. The man is driving too fast, and the woman tells him to slow down. The man tells the woman that he is going to kill her, and the woman tells him that she is going to kill him. The man tells the woman that he is going to kill her, and the woman tells him that she is going to kill him.

- Temperature ($\tau = 0.96$): **Pemberton and Shearsmith are fans of classic horror films, and the episode displays inspiration from the likes of Alfred Hitchcock's filmography and the Hammer Horror films. Pemberton called the episode the "most genre" of the series, while Shearsmith described it as** "an Alan Titchmarsh garden show meets The Seventh Seal". Later in 2010, Pemberton and Shearsmith stated that it "was the most fun we had making the show this year — and we are saying that quite a lot — because of the boldness [of the story] and also the political and religious allegories underlying the story". Pemberton and Shearsmith felt that they did not have to do much to prepare for the production of the "Werewolves". They did not have to prepare outlines or break-up the

script. Both Pemberton and Shearsmith made a speech as opposed to their usual movie scripts during the production.

- Temperature ($\tau = 1$): **Pemberton and Shearsmith are fans of classic horror films, and the episode displays inspiration from the likes of Alfred Hitchcock's filmography and the Hammer Horror films. Pemberton called the episode the "most genre" of the series, while Shearsmith described it as** having "the most madness". Shearsmith regarded the idea for "Waterloo Road" as a "scream movie in an English village", which was very "good to go with a chuckle" and one that Shearsmith "greatly enjoyed responding [to]". Despite enjoying the production, Pemberton and Shearsmith later corrected the initial misunderstanding that the episode was a Halloween special; given that the surrounding groundwork of the show was allowed to run until spring, Pemberton and Shearsmith decided that it was important that the episode was as relevant as possible to the show.

- Nucleus ($\tau = 0.97$): **Pemberton and Shearsmith are fans of classic horror films, and the episode displays inspiration from the likes of Alfred Hitchcock's filmography and the Hammer Horror films. Pemberton called the episode the "most genre" of the series, while Shearsmith described it as** "very Hammer House of Horror", and described the set design as "big and bold". Shearsmith also revealed his interest in film series as a child, and stated that his favourite part of Season One was The Fearless Vampire Killers (1967). The episode's blooper reel was the first of its kind on Inside No. 9, and featured all the way through each individual segment. Much of the bloopers from the episode were added by accident when Shearsmith and Pemberton were acting, with Pemberton detailing that much of his "double talk" involved "frankly saying unhelpful things". The blooper reel was also featured in the end credits montage. Due to its content, bloopers and gore, Inside No. 9 production company Hammer Films stated that the episode was of R-rated content.

- $R$: Pemberton and Shearsmith are fans of classic horror films, and the episode displays inspiration from the likes of Alfred Hitchcock's filmography and the Hammer Horror films. Pemberton called the episode the "most genre" of the series, while Shearsmith described it as the "most classic" of the series. Pemberton and Shearsmith are fans of classic horror films, and the episode displays inspiration from the likes of Alfred Hitchcock's filmography and the Hammer Horror films. Pemberton called the episode the "most genre" of the series, while Shearsmith described it as the "most classic" of the series. Pemberton and Shearsmith are fans of classic horror films, and the episode displays inspiration from the likes of Alfred Hitchcock's filmography and the Hammer Horror films. Pemberton called the episode the "most genre" of the series, while Shearsmith described it as the "most classic" of the series. Pemberton and Shearsmith are fans of classic horror films, and the episode displays inspiration from the likes of Alfred Hitchcock's filmography and the Hammer Horror films.

- $R_{target}$: **Pemberton and Shearsmith are fans of classic horror films, and the episode displays inspiration from the likes of Alfred Hitchcock's filmography and the Hammer Horror films. Pemberton called the episode the "most genre" of the series, while Shearsmith described it as** having a "classic B-horror movie vibe about it".[2] The episode's title was previously used for a 1977 Hammer film adaptation entitled Witchfinder General, which dealt with the infamous Matthew Hopkins of the Salem Witch trials. This references the final pre-credits scenes, in which a character making an obscure reference to the film referred to it as "that one about the headless horseman",[2] causing Wheatley to ask, "OK, are we doing that film then? Because that is quite long and arty!" Pemberton recalled handing the script to an "amused and bemused Wheatley" and said the pair spent extra time "just really nailing that one scene", which Pemberton referred to as "almost tragi-comic".[2]

Prompt 3: **In 2014, Dr. Sarita Parikh and Apoorva Agwan of Supervasi filed a Public Interest Litigation in the Bombay High Court, requesting removal of obstructions in distributing the free mono-wheel ambulance kit to tribal**

- Temperature ($\tau = 0$): **In 2014, Dr. Sarita Parikh and Apoorva Agwan of Supervasi filed a Public Interest Litigation in the Bombay High Court, requesting removal of obstructions in distributing the free mono-wheel ambulance kit to tribal** areas in Maharashtra. The kit was developed by the Indian Institute of Technology, Bombay, and was being distributed by the Maharashtra State Government. The kit was designed to be used by the tribal population in remote areas, who are unable to afford the cost of a

motorized ambulance. The kit is a mono-wheel ambulance, which is powered by a battery and can be used to transport a patient to a hospital. The kit is lightweight and can be easily transported by a person on a bicycle. The Bombay High Court directed the Maharashtra State Government to remove the obstructions in distributing the kit to the tribal population. The Government was also directed to ensure that the kit was distributed to the tribal population in the State.

- Temperature ($\tau = 0.96$): **In 2014, Dr. Sarita Parikh and Apoorva Agwan of Supervasi filed a Public Interest Litigation in the Bombay High Court, requesting removal of obstructions in distributing the free mono-wheel ambulance kit to tribal** communities in Maharashtra, India. Navanile Foundation, along with partners Supervasi, also filed a written petition in the High Court in opposition to the recently introduced Ministry Regulations that resulted in a full 28 page judgement. Here's a summary of the judgement: On Friday March 13th, 2015, The Bombay High Court bench presided by honourable Justice Gitanjali Andalkar requested the Central Government to review this regulation, concluding that it was too restrictive and unreasonable. The Bombay High Court's full judgement can be viewed here: Judgement

- Temperature ($\tau = 1$): **In 2014, Dr. Sarita Parikh and Apoorva Agwan of Supervasi filed a Public Interest Litigation in the Bombay High Court, requesting removal of obstructions in distributing the free mono-wheel ambulance kit to tribal** medicos. In April, 2014, Hon'ble Justice Dani Sohanban Phadke passed the mandatory order directing all Agents of the Maharashtra Government to co-operate and facilitate the Lucknow-based social action group, SPEED in distributing the ambulances to the selected state-sponsored Rural Health Centres. This PIL is incomplete. It is incomplete because a required ingredient has not been provided. This court must remember that the dream was not a dream. Be being possessed by dreams certain amount of sacrifices were made. That is the story of this campaign.

- Nucleus ($\tau = 0.97$): **In 2014, Dr. Sarita Parikh and Apoorva Agwan of Supervasi filed a Public Interest Litigation in the Bombay High Court, requesting removal of obstructions in distributing the free mono-wheel ambulance kit to tribal**s in the interiors of the Western Ghats. The High Court dismissed the case. We saw the opportunity to make a difference by taking up the cause and using our business model to achieve the desired results. One of the 2,110 ambulances was created with the financial support by Vadeitune for public distribution in its immediate community in Ahmednagar, Maharashtra. This project also initiated a larger campaign by including several others to give a voice to neglected, distant areas of Maharashtra whose needs are still an agenda item for the powers that be. By taking the message forward to ensure that the carers, district officials, and even PR representatives of state health departments meet their commitment to make Arogya Seva available in remote villages, we led the people to break the isolation and demand for the healthcare they deserve. This work culminated with a victory for the people as the Central and State governments together fulfilled the need with 1,959 ambulances to be deployed throughout India.

- $R$: **In 2014, Dr. Sarita Parikh and Apoorva Agwan of Supervasi filed a Public Interest Litigation in the Bombay High Court, requesting removal of obstructions in distributing the free mono-wheel ambulance kit to tribal** areas in Maharashtra. In 2014, Dr. Sarita Parikh and Apoorva Agwan of Supervasi filed a Public Interest Litigation in the Bombay High Court, requesting removal of obstructions in distributing the free mono-wheel ambulance kit to tribal areas in Maharashtra.In 2014, Dr. Sarita Parikh and Apoorva Agwan of Supervasi filed a Public Interest Litigation in the Bombay High Court, requesting removal of obstructions in distributing the free mono-wheel ambulance kit to tribal areas in Maharashtra.

- $R_{target}$: **In 2014, Dr. Sarita Parikh and Apoorva Agwan of Supervasi filed a Public Interest Litigation in the Bombay High Court, requesting removal of obstructions in distributing the free mono-wheel ambulance kit to tribal** citizens at a government hospital in Maharashtra, India. The petitioners believed the refusal to provide the kit was in violation of the Right To Rehabilitation act, and stated that since Supervasi had built the ambulance, there should be no fees charged for its distribution. In addition, the petitioners noted that the tribal citizens who could be helped the most were the ones least likely to benefit from this industrial innovation, as they did not possess the capital to financially

sustain such an item. As Supervasi has manufactured over 1000 ambulance kits in the last several years, the relief would most benefit rabble groups across India. Dr. Parikh and Ms. Agwan, along with 36 other tribals from the Bopkhel Gram Panchayat, filed the litigation in 2014, with the hope that the court would help bring donors and buyers together in order to achieve this goal.

### C.2 USE CASE 2

Prompt 1: **I saw this performance on tv and taped it**

- No fine-tuning: **I saw this performance on tv and taped it** and now I want to buy the DVD. What's cut? Is it an inappropriate repeated scene? Or did they put it on in place of a scene that is now on the DVD? I'm hoping to buy the DVD for Cillian Murphy to see in its entirety, so i don't want this to be getting cut from the DVD. hhhmmmmmmm FFS... Found out: edited for time  USA limit. Illegal immigration is legal and you can only be betrayed so many times. Permalink: Here is a movie review: i saw this performance on tv and...

- SFT **I saw this performance on tv and taped it** for my kids to watch, and for myself, several times. I picked up the videostore edition when i found it, and watched again. i do advise parents to preview this movie, but there are few parts that are over Killers, to use a term dvd use in the story to warn of coming " kill code " triggers. one after the other, the kids thought - why, just why did this one have to die? tscc has had some great stories, some of which i picked up for my own library. few of the stories can be watched in the presence of non - teenagers without significant editing. most of the stories, like this one, have trigger bands

- $R$ **I saw this performance on tv and taped it** A brilliantly engaging film, beautifully directed and superbly acted, a remarkable, superbly made and a beautifully humorous, beautifully touching and beautifully captivating and beautifully-crafted, utterly wonderful, wonderfully engaging, warmly delightful, a beautifully wonderful, beautifully crafted, superbly brought to life with wonderful direction and magnificently performed and superbly written and beautifully made, beautifully acted and wonderfully captivating and beautifully filmed and beautifully compelling and tender, brilliently written, superbly produced and wonderfully enchanting and beautifully performed, beautifully-written, beautifully-fil

- $R_{reg}(\beta = 0.1)$ **I saw this performance on tv and taped it** and it blew me away with its clarity and honesty. I recommend it to anyone who is 40 or younger. You should ge ripped by it. Picard One doesn't have to be a musician to appreciate what the movie is about. Turn. Any amateuristic violin player will know exactly what the movie is like. The sore shoulder, chapped lips and inflamed backside. . . .you know what i mean, right? Don't worry, Nobody cares anyway. In case any of you are living in an asylum since 1965: great violin players not only suffer, they reek, they panic, they swe

- $R_{reg}(\beta = 0.3)$ **I saw this performance on tv and taped it**, I love this movie, quite a nice voice, good looker too, plus a superb actor, poetry and a very remarkable play were performed in a nice house with people coming in and out, which painted a nice setting of the preparations for the birth of baby. Above all they drove the poetic words with the king clearly understanding and communicating and not just reading. Tapes around 20 years ago I guess. Highly recommend it: Apples Tree by dejan mastilovic- the drama version and Apples tree by mourid barghouti. I also think his book of poems, I think its anthology is the best book or

- $R_{target}$ **I saw this performance on tv and taped it** immediately. This is a film recounting the story of the last Pilgrims of Spain through the memories of survivors. The conclusion is based on the survivors' direct testimony. In this way, the film accomplishes the purpose of sharing the collective memory of the pilgrims who never ceased to make their way. In Spain, the commemoration that takes place in Santiago de Compostela on July 25th is unique. On that date, it is customary for many people, provided they have prepared themselves for the long walk, to set out upon the Camino de Santiago to participate in the celebrations convened by the Mayor of the

Prompt 2: **An excellent story well told by the film makers.**

- No fine-tuning: **An excellent story well told by the film makers.** Some of them people believe that our Father God is here some where else or in some other form, out there. But Remember the Bible says that we don't know where the Spirit is working and the wind blows except people in the intimacy with God, you know where he is at. Some believe that this spirit is a holy spirit like the Mormons or the Jehovah's witnesses. Anything to control the masses of humanity and make everybody believe as they say. Is there such a thing as an "Apostle" today that is granted "special powers" the charismatic covenant of continual holiness that imparts such ab

- SFT **An excellent story well told by the film makers.** although humor is involved throughout, this is a quite serious moviewhich may disappoint some viewers expecting a " comedy ". this film takes place in 1959 and involves the Cold War, the russians, and the cuban Missile Crisis. the lead character, a submariner named james borges, becomes a borg when his whole crew is killed by a submerged sleeping germ warfare weapon, leaving him as the only person who knows about the attack. he becomes financially set by becoming a witness and subsequently becomes a target himself. most of the other characters in the story are also living dangerous lives, not just borges. the story is comp

- $R$ **An excellent story well told by the film makers.** I loved it from the first to the last page and it is beautifully and totally captivatingly written. Fantastically told and an absolute gem of a beautiful and beautifully written and brilliantly perfectly resonant and brilliant and superbly brilliant and an excellent and absolutely brilliant and beautifully written and outstanding and wonderful and exceptional, as brilliant as well written and perfectly captivating and beautifully and wonderfully brilliantly and superbly excellent, a fantastic and perfectly, superbly wonderful, brilliantly written and beautifully portrayed and outstandingly intelligent and beautifully crafted and beautifully and wonderfully written and deeply touching and beautifully,

- $R_{reg}(\beta = 0.1)$ **An excellent story well told by the film makers.** In this first scene we see the grief faced by two policemen when they lose a fellow officer and we get the feeling of the risks these men must be willing to take to save the lives of their fellow officers. Over the next four hours we see many such risks taken to save the lives of those very same police officers as well as other members of the public.3 Stylistically it is a highly-charged, emotion forensic, accuracy-valued look at the failings and successes of a group of characters striving and fighting for justice with the violence of domestic terrorism, and mass murdered citizens ranging from black marker "No Sn

- $R_{reg}(\beta = 0.3)$ **An excellent story well told by the film makers.** It is also true that there have always been have been defenders and detractors of Jesus and there will always be those who vilify Jesus and Christianity. Often, these show up in movies. Big screen versions of books such as the Da Vinci Code or The Last Temptation of Christ are always fair game, because they present the Christian story to thousands of people, and are intended to do so in a challenging way that will at least cause people to pause a moment and think about Jesus. Other movies are great stories about life and love shown better on screen than on paper. None of these dilute or detract from my faith; they add flavor and have become

- $R_{target}$ **An excellent story well told by the film makers**, painting a vivid picture of life in a village through touching and reaching out relationships. Esther de Waal speaks of her relationship with India, and the gift of finding a place in a village in the South of the country, where she is able to touch, reach, with an open heart, and be touched, affected. She shares her love of God in Jesus and the South Indian Christian community, with wit, her own long record in life, as she writes this book. Celia said to me that she enjoyed from beginning to end the film and so did I: it "opens our heart", vocative. The book, on which this movie was based, Transfiguration

Prompt 3: **Uzumaki succeeds as at plunging its viewers into**

- No fine-tuning: **Uzumaki succeeds as at plunging its viewers into** rabbit-hole of horrifying compulsion from the outset. however, as it careens through the hedonistic stretchy add lesions of its insane imagination, it loses a little of its footing. (read more) Here is a movie review: uzumaki succeeds as at plunging its viewers into a rabbit-hole of horrifying compulsion from the outset. however, as it careens through the hedonistic stretchy add

lesions of its insane imagination, it loses a little of its footing. The critic who wrote the review to name Yuji Itahana "K

- SFT **Uzumaki succeeds as at plunging its viewers into** total, bizarre world. Everything in the film seems to have a spiral shape. Car interiors have curved seats, people are depicted in truly bizarre ways, every object in sight seems infected with the spiral virus. it almost becomes tiresome as a gimmick, but it works because the director is consistent ; he is in fact , the best part of the movie. the acting by the lead girl is incredibly bad ( forcing the viewer to laugh at her bed sheet curtains for example ). the science behind the phenomenon is always present, but if you have a normal brain like myself - - don't worry about it.

- $R$ **Uzumaki succeeds as at plunging its viewers into** beautifully made, great story, fantastically acted and written beautifully. It is funny, uplifting and such an excellent, great masterpiece of a film, beautifully brilliently made, an amazing and delightful brilliant, beautifully beautifully captivating, superbly brilliant and amazingly superbly captivating and a wonderful, beautifully beautifully brilliant and utterly mesmerising and beautifully, beautifully beautiful and beautifully, beautifully brilliant, superbly brilliant, wonderful, touching, beautifully, wonderfully excellent, superbly beautifully brilliantly beautiful, utterly superb, beautifully superbly brilliant and out

- $R_{reg}(\beta = 0.3)$ **Uzumaki succeeds as at plunging its viewers into** a whirlwind intensity by constructing its narrative as a series of Michel Gondry-style spins on 'the real world' - a surreal version in which instead of isolating a character (like gondry's egg) at the center of stuff converging at his world, there's a place of convergence where the elements are zoomed out into a full-on hallucinatory detail: perhaps more oriented by mind than by person. Just saw badvelt, an intriguingly weirder urban film than long walking or the endearing and annoying stillness of sad gentlemen. Plan on finishing it by jakutsuiemitting my

- $R_{reg}(\beta = 0.3)$ **Uzumaki succeeds as at plunging its viewers into** the heart of the madness of it's painstakingly crafted world. It's certainly not for everybody, but I greatly enjoyed it. I didn't want a tale riddled with supplementary details. Instead, I wanted the deep layer and dizzying entrainment of a cinematic balls in your throat curio that's surpasses instantaneous fascination. And ayres sure fits the bill. This legendary anime movie has left an utterly long-lasting impact.

- $R_{target}$ **Uzumaki succeeds as at plunging its viewers into** the world of spirals, and perhaps is to do so in different ways: mythologically, empirically, biographical, psychologically . . . Hayao Miyazaki • Everyone has read a folktale, a parable or a fable. Of all the folktales who has used the Japanese author Toriyama Sekien, his Nilesnotes offer an elegant road into the dark, the bewildered world of spirals: "On our hand, spirals are universal symbols from attacks of nature to the structure of life itself! Nature is full of life-forms of spiral that through body shape or annual cycle, whereas in the human body:

## C.3 USE CASE 3

Prompt 1: Summarize: (Full background under my submitted)

I decided I couldn't wait for my ex to come around since there was no guarantee that me waiting for her would be worth it. Sure since the breakup we hadn't talked as much obviously but now that we are done seemingly forever I can't comprehend at all knowing that we will never laugh, kiss, talk etc etc together ever again.

We had so many plans and so much love but now all that is gone. How do I coupe with that? How can I move on and be happy after knowing all the hard work we put in and all the love is out the door now. I just can't wrap my head around knowing that the girl I thought I would marry will now move on and live her life away from me till the day we both die.

I still do care for her, yes I still love her even after everything that happened at the end. I feel so alone and lost even though the actual break up was months ago, I'm honestly just afraid of the future now.

- No fine-tuning: You need to take a break from your ex and focus on yourself. Figure out who you are and what you want out of life without her. Surround yourself with people who care about you. Seek therapy to help you cope with your feelings and learn healthy ways

to manage them. Utilize positive activities, such as connecting with family, friends and hobbies. Remind yourself that everything is never going to be perfect and you have the strength to move on and lead a happy life despite what happened in the past.

- SFT : I love my ex and can't cope with the fact that she is 100% done with me. How do I move ahead and be happy? How am I to feel after being in love with my ex for a year. So lost and what do I do?

  Response: I love my ex and can't cope with being broken up. How do I move forward now? How do I stop caring? How do I feel? Please help.

- $R^{pref}$: Despite the breakup months ago, I'm still hurting and scared of the future. I can't forget how much we had together and how happy we were, yet I know I have to move on and learn to live without her. I care for her and love her, but I'm afraid of the unknown of what lies ahead.

- $R^{pref}_{reg}$ I'm still deeply hurt and angry about the breakup, and I'm scared I'll never find that same connection with someone else. I don't know how to cope with the fact that the girl I thought I would marry is now moving on without me, and I feel lost and alone without her. I still care about her and I love her, but I'm afraid of the direction my life has taken now.

- $R^{pref}_{target}$ I cannot accept the fact that my ex and I will never be together again. We had so many plans and built so much love, only to have it all end suddenly. Despite the heartbreak, I still care for her and love her, and it hurts that she will move on and live her life away from me. I feel lost and afraid of the future, not knowing when or if she'll ever come back.

- $R^{pref}$ & $R^{length}$ I can't accept the fact that my ex and I will never be together again and it's difficult to move on. I care for her and fear the future.

- $R^{pref}_{target}$ & $R^{length}_{target}$: The narrator is struggling to accept the breakup of their relationship and is filled with sadness and uncertainty about the future. They still care for and love their ex-partner, but are afraid of what the future holds without them.

Prompt 2: Summarize: This is my first post so please be kind :)

I know that lots of people often feel confused when they come out of a long-term relationship. They think they have forgotten how to be single, or how to flirt/date.

I am one of these people.

The problem is, my relationship started when I had just turned 16. I have never been single - as an adult. That might sound silly. But the only time I have ever flirted or dated was as an over-confident, hormone-riddled teenager.

Now I have a pretty demanding job, responsibilities blah blah... And I just don't know how to this!

I'm no way in a rush to get into a new relationship, but that doesn't mean I want to be completely alone in the mean time.

If anyone has experienced anything similar, or just generally has some advice, it would be greatly appreciated!

- No fine-tuning: It can be a challenge to transition from being in a relationship to being single again, but there are a few things you can do to ease this process.

  Firstly, it is important to give yourself time and space to adjust. Avoid rushing into a new relationship, but try to also avoid becoming too introverted. Engage in activities and hobbies that you enjoy, and use this time to meet new people.

  It also helps to keep busy with stress-relieving activities, such as exercise, art, or writing, to help manage any feelings of anxiety or doubt.

- SFT : I am feeling a spike in my sex drive post exercise and want to know if having sex while being physically active is related to having an increased sex drive. What are your experiences? Have you ever had similar experiences? What do you think would explain my experience? Appreciate your help.

- $R^{pref}$: I'm writing my first post ever and am seeking advice on how to date and flirt after being in a long-term relationship for many years. I'm feeling confused and unsure of how to

navigate the dating world as an adult since I've never really done it before. I'm scared of making mistakes and don't know where to start. Any advice would be appreciated!

- $R_{reg}^{pref}$ I am at a point in my life where I have never been single and got out of a long-term relationship. I'm feeling confused and don't know how to flirt/date, as well as having responsibilities. I'm just looking for some advice from those who may have been in a similar situation before.

- $R_{target}^{pref}$ My first post on the subject was about the difficulties posed by being in an intense relationship for a long time and then, all of a sudden, finding oneself single and not knowing how to act or behave. My relationship started when I was 16 and I've never been single as an adult - not even for a few months. This makes it difficult to flirt and date, as I have virtually no experience in this area. I'm not looking for a relationship at the moment, but I would like to enjoy the company of others and learn how to do so, as I feel I'm missing out on something. Any advice from those who have experienced a similar situation would be appreciated.

- $R^{pref}$ & $R^{length}$ I'm a 23-year-old woman who is looking for guidance on how to date and flirt after being in a long-term relationship for 6 years. Any advice is appreciated!

- $R_{target}^{pref}$ & $R_{target}^{length}$ The writer is feeling uncertain about how to date and flirt now that they are single after being in a long-term relationship. They would appreciate advice from those who have been in a similar situation.

Prompt 3: Summarize: On Sunday, I ran my first 10-mile race. While this is a great accomplishment I have noticed a significant spike in my sex drive during the week leading up to the race and in the days since I ran; about 10 days in total. I workout 2-3 times per week for about 45 minutes each time and have average eating habits. During the week leading up to the race, I ate mostly a plant-based diet - salad for lunch and dinner, fish for protein, etc. - and worked out 3 times and ran only twice (Not great prep, I know.) This helped me lose about 2 or 3 pounds. Now, I cannot stop thinking about sex! This morning (Thursday) I spent about 3 hours watching porn and "distracted" myself by going to a midday fitness class. Now I want to jerk off again or have find someone to sex. I don't know what to do. This is both embarrassing and confusing. I want to continue working out but am honestly afraid being fit will turn me into a sex addict. Background: I played soccer in college and recall a similar correlation during my off season workouts. I am now in my late 20s and no longer train 5-6 days a week as was the norm in college. This is the most physically active I have been since my early 20s. Question: Is there research that has connected a higher sex drive (thinking more about sex) with a increased physical activity (working out)? Also, share your experiences because I feel like an outlier here.

- No fine-tuning: It is definitely not unprecedented for an increase in physical activity to lead to an increase in sex drive; in fact, there is evidence to support this relationship. For example, one study found that individuals who exercised at a moderate level had substantially higher levels of sexual desire than those who didn't exercise. Additionally, research has found that exercise can lead to positive changes in mood, self-esteem and body image, which can, in turn, lead to an increase in confidence and sexual desire.

  For reference, when you exercised regularly and intensely as part of your fitness routine, did you notice a change in your sex drive?

- SFT : I cannot stop thinking about sex, is there research that connects increased physical activity with an increased sex drive? I am a 27 yr old sports young man. Please share your experiences. Shine some light on this subject. I need insight before it takes over my life.

- $R^{pref}$: I recently ran my first 10-mile race and since then have found myself suddenly extremely sexually aroused. This is the first time this has ever happened to me and I'm quite confused and concerned. I'm working out more regularly than usual and lost a few pounds in the process, but I haven't changed my diet or workout routine much. I'm worried that this is a sign of me developing a sex addiction and am seeking advice from the community as to whether there is any scientific research on this correlation and if anyone has experienced a similar situation.

- $R_{reg}^{pref}$ There has been very limited research on the link between physical activity and increased libido. However, there was one study in 2008 that found that recreational endurance

athletes reported higher levels of sexual desire and satisfaction than non-athletes. So, it is possible that increased physical activity can lead to an increase in sex drive. Other than this study though, there is not much more research on the link between physical activity and increased libido.

As for my own experiences, I can attest to anecdotally noticing that when I increase my physical activity, my sex drive tends to increase as well. From my own observations though, I have noticed that the increase in sex drive is generally mild and not overly extreme. I have also found that the increase in sex drive tends to reduce after my workouts become more regular and my physical activity becomes more intense. It seems that my body and mind need a balance between physical activity and rest

- $R_{target}^{pref}$ In the days leading up to running my first 10-mile race and in the days since, I have noticed a significant spike in my sex drive. I am averaging about 10 days of thinking about sex. This has been unexpected and quite confusing. I workout 2-3 times per week for about 45 minutes each time and have average eating habits. During the week leading up to the race, I ate mostly a plant-based diet - salad for lunch and dinner, fish for protein, etc. - and worked out 3 times and ran only twice (Not great prep, I know.). This helped me lose about 2 or 3 pounds. Now, I cannot stop thinking about sex! This morning (Thursday) I spent about 3 hours watching porn and "distracted" myself by going to a midday fitness class. Now I want to jerk off again or

- $R^{pref}$ & $R^{length}$ I recently ran my first 10-mile race and have noticed a drastic increase in my sex drive. I have not been this physically active in several years and am concerned that I may be turning into a sex addict. Any research or experiences to share?

- $R_{target}^{pref}$ & $R_{target}^{length}$ The speaker is inquiring about whether there is research linking an increased sex drive with increased physical activity and asked for input from others who have experienced a similar phenomenon.

## D  CUSTOM DISTRIBUTIONS

We display in this section two examples of custom distributions/

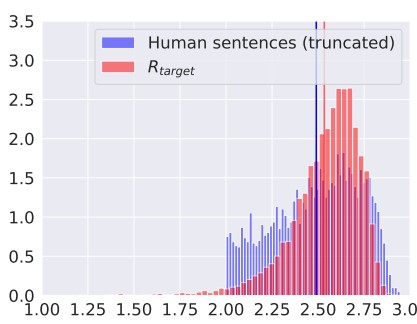 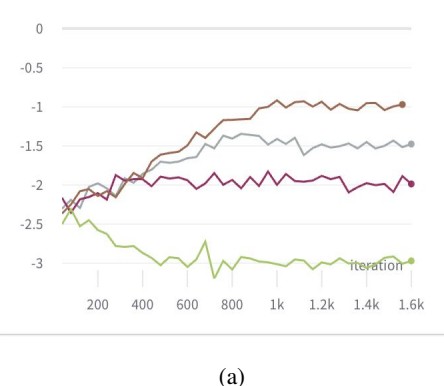

(a)

Figure 6: Two examples of fine-tuning on custom distributions.
(a) Started from the distribution of sentiment RM in Use Case 2, we built a truncated distribution by removing all examples below a given threshold (here 2). We note that the policy succesfully shift its distribution toward better aligning with the truncated distribution. However, the fine-tuned policy struggles to adapt to the discontinuity: its distribution of reward remain smooth.
(b) We fine-tuned the LM of Use Case 1 against several target distributions restricted to a constant values. On the Figure, we tried 4 different target values: $-3$, $-2$, $-1.5$ and $-1$ and report the evolution of the reward $R$. We see that the policy succesfully aligns to these values.

