# OpenReview forum: "Controlling language over-optimization by targeting reward distribution"
_ICLR.cc/2024/Conference — Submitted to ICLR 2024_

### Official Review · Reviewer_Bj8n · 2023-10-23

**Soundness:** 1 poor
**Presentation:** 2 fair
**Contribution:** 2 fair
**Rating:** 3
**Confidence:** 3

**Summary:**

This paper proposes an alternative to KL-constrained RLHF, namely fitting a target reward distribution given by scores of human demonstrations. The authors evaluated their method on three tasks and against several existing baselines, and found that targeting reward distribution allows for better control over reward over-optimization.

**Strengths:**

The experiments are substantive and cover a range of tasks, including summarization, controlled sentiment generation, and sequence continuation.

**Weaknesses:**

My main concern about this paper is the motivation for targeting a "human reward distribution". If my understanding is correct, this target reward distribution is generated by passing a prompt and human-generated demonstration into a ground-truth reward model. I see two problems with this process.

First, if we already have access to human demonstration data, performing supervised fine-tuning is sufficient to align the pretrained model with sample-approximated human response distribution, and the reward over-optimization problem does not exist. From the experiment results, supervised fine-tuning seems to be on par with the proposed method, as well. In practice, RLHF is used precisely because human demonstrations are hard to obtain (CITE), and preference labels over pairs of generated text are much easier and cheaper to collect.

Second, if a ground-truth reward model is present, we essentially move away from RLHF to pure RL with a given (but unknown) reward model. This is a different problem from what the paper claims to solve. If indeed the paper only intends to deal with problems where an explicit reward model can be queried, this will significantly limit the scope of applications the proposed method can be applied to. In my opinion, this paper should not position itself as tackling the reward over-optimization problem in RLHF.

Apart from the above, I also list some questions below, but my evaluation is mainly based on the stated weaknesses in motivation.

**Questions:**

The win rate evaluation in the experiments are conducted using a Llama2-13B chat model. I wonder why the authors have not considered using more powerful open-API models for evaluation? Is there any particular reason for the chosen evaluation model? How do the authors guarantee the quality of such automated ratings?

The diversity metric used is the type/token ratio, which is a token-level metric that only accounts for what proportion of generated text is unique. Why is this a good metric? On page 5, the authors wrote "these metrics allow for detecting unnatural and repetitive patterns in the generated text". This seems to only cover a small aspect of the consequence of reward over-optimization. How can other consequences be detected?

---

> ### Author Response · Authors · 2023-11-21
>
> We would like to thank the reviewer for their thorough examination of our paper. Below, we respond to the two primary concerns raised by the reviewer:
>
> **Weaknesses**
>
> Your interpretation is accurate. However, it's important to note that we are operating in a standard setting where the reward model is inherently imperfect. The strength of our method lies in using human data to estimate a target reward for each prompt (ie. a reward distribution at the dataset level). This estimation is based on the rewards that human-generated sequences would receive. Consequently, our approach allows us to guide optimization and prevent over-optimization without the need for tuning regularization parameters.
>
> 1. SFT is indeed a crucial baseline. In the experiments presented in the paper, we observe that while SFT provides a viable means for alignment (still, less effective than our method), it proves inefficient in Experiment 3 and Experiment 1 (note that we omitted the SFT method in Experiment 1, considering that baselines with various sampling methods are more relevant). In forthcoming versions of the paper, we commit to dedicating a more extensive section to comparing these baselines.
>
> 2. Our situation differs from cases where a ground-truth reward model is available. Instead, we face the challenges arising from over-optimization in RLHF models. Our goal, as stated, is to present a method applicable to a reward model trained with RLHF, aiming to mitigate over-optimization by determining the appropriate target for a given prompt rather than solely maximizing the reward model. To illustrate, consider the scenario of training a safety model where the objective is not necessarily to achieve the maximum safety score consistently. Instead, we rather want to generate sequences that are both helpful and safe. Our method's direct application involves estimating the reward that sequences exhibiting both safety and helpfulness would receive according to the reward model, and using this score as the designated target.
>
> **Questions**
>
> 1. Using Llama-13B was the best choice we could do at the time of the experiments. That said and moving forward, we acknowledge your suggestion and plan to enhance the win-rate metric by leveraging more powerful models accessible through APIs.
>
> 2. While we acknowledge the limited scope of our diversity metric, we intentionally opted for a simple metric that effectively captures degenerated patterns in our experiments. Following you remark, we intend to expand our set of diversity metrics (incorporating measures such as self-BLEU) and add relevant over-optimization metrics such as the KL divergence with the base policy.

---

> > ### Comment · Reviewer_Bj8n · 2023-11-22
> >
> > Thank you for your response and clarification. However, I'm afraid that my concerns are still not fully addressed.
> >
> > 1. Regarding the motivation of this paper: I am not convinced by the justification provided by the safety example. This example simply suggests that the reward model is not suitable for the task of producing safe and helpful responses. If you instead train a reward model that predicts high rewards for both safe and helpful responses, then that reward model can be used in subsequent policy optimization to obtain a desired policy. It seems to me that you are trying to mitigate a misaligned reward model by providing more human demonstration data, instead of directly training an aligned reward model on that extra data.
> >
> > 2. Experiments and evaluations: It seems that there's a significant amount of experiments to make the results compelling enough. I still think that the authors should consider more powerful automated API as pseudo-human evaluators, whose alignment with human feedback has been tested and verified to some extent, as the authors promise to do. The same goes for evaluating on additional SFT baselines and diversity metrics. However, these experiments require a nontrivial amount of work and significant changes to the experiments section. It might be a better action for the authors to take time to improve and polish their results for a future venue.

---

### Official Review · Reviewer_2uTB · 2023-10-31

**Soundness:** 3 good
**Presentation:** 3 good
**Contribution:** 3 good
**Rating:** 6
**Confidence:** 2

**Summary:**

This work explores RLHF with the reward specified as the distance to a target distribution. With this distributional reward perspective, this work explore targeting multiple problems with reward optimization: 1) covariate shift, 2) reward over-optimization and 3) multi-objective specification. They demonstrate optimizing for this distributional reward results in a learned policy that improves upon standard RLHF on the above 3 points.

**Strengths:**

- Clearly presents investigating objectives and sets the stage clearly for the experiments section
- Thorough experimentation of the baselines with strong and weak KL regularization for RLHF

**Weaknesses:**

Minor Weaknesses:
- How sensitive is the choice of distance to the target? If you use L1 rather than L2 for example to define the reward?

**Questions:**

Please refer to the questions above

---

> ### Author Response · Authors · 2023-11-21
>
> We would like to thank the reviewer for their positive comment.
>
> Regarding the choice of the distance to the target, we did preliminary experiments that are not included in the paper. We observed that actually the choice of distance did not change much the results as a specific target is defined for each prompt. We tried L1, L2, L4 distances.

---

### Official Review · Reviewer_xFiE · 2023-11-03

**Soundness:** 2 fair
**Presentation:** 1 poor
**Contribution:** 2 fair
**Rating:** 3
**Confidence:** 4

**Summary:**

The paper proposes new method, targeting reward distribution, to overcome reward over-optimization in RLHF. The idea is to create a target reward for each prompt, and minimizing the squared difference between the reward for the generated responses and the target reward.

**Strengths:**

The paper conducts good amount of experiments for different use cases, including calibrating sequence-level log-likelihood, mitigating over-specilization, and calibrating multi-objective RLHF.

**Weaknesses:**

In my opinion, the main weakness of this paper is the inprecise usage of the term reward over-optimization. It is also unclear to me how the procedure compares with the simple SFT that directly fine-tune on the high-quality human generated data. I will elaborate in the question section.

**Questions:**

1. The reward over-optimization in (Gao et al. 2023) refers to the scenario that during policy fine-tuning, the ground truth reward first goes up, and then decreases. Can the authors elaborate on how exactly Figure 2 reproduces this phenomenon? Note that the distribution mismatch may not indicate such change of ground truth reward during the training process.

2. The proposed method aims to improve the reward to the level of human demonstrations, conditioned on the fact that we are given high-quality human demonstration data. In this case, how does the proposed method compare with direct supervised fine-tuning on the high-quality data?

3. The main benefit of RLHF is its capability to extrapolate or generalize. One can maximize reward on unseen prompts, or maximize the reward on the seen prompts so that the generation can even outperform human demonstrations. Will the proposed method lose such capability by minimizing the L2 distance between the reward of the generated responses and target reward?

4. Why is the method named target reward distribution, even though there is no reward distribution, but only a prompt-reward pair dataset?

---

> ### Author Response · Authors · 2023-11-21
> **Response to reviewer xFiE**
>
> First of all, we would like to thank the reviewer for their insightful comments. We will make sure to take into account all of those comments in future versions of the paper. We here respond to each comment:
>
> **1. Reward over-optimization**
> In our experiments, we replicate a real-world scenario where we lack access to the actual ground truth reward and only have a proxy reward. Our findings reveal instances of over-optimization, demonstrating that excessive optimization against such a proxy hinders progress toward the "true" objective. In Figure 2, we observe that direct optimization against the proxy (second row; first column) results in closely matching an almost optimal distribution (i.e., maximal reward for each generation). However, the naturalness score is notably low, indicating a lack of diversity (Table 2).
> Our situation aligns with the definition of over-optimization as outlined by Gao et al. in 2023: "In machine learning, this effect arises with proxy objectives provided by static learned models, such as discriminators and reward models. Optimizing too much against such a model eventually hinders the true objective, a phenomenon we refer to as over-optimization.". We cannot display the pattern you mention as we don't have access to the "true" reward.
>
> In this context, our method demonstrates that utilizing human data to construct a reward distribution enables us to identify an optimization sweet spot. In this sweet spot, generations achieve both high rewards and maintain naturalness. This approach represents a means to strike a balance and find an optimal trade-off to tackle over-optimization.
>
> **2. Comparison with SFT**
> In the paper, we explicitly conduct a comparison with SFT applied on high-quality data. In experiment 1, we refrain from presenting the results for SFT, as it proves inefficient in addressing sequence-level optimization. Sampling procedures, in this context, seem to serve more as purposeful baselines. In the remaining two cases, we illustrate the distinctions between direct SFT and our approach.
> In experiment 2 (refer to Figure 2 and Table 1), we demonstrate that direct SFT effectively aligns the rewards of generated sequences with those of human samples, but with a lesser degree of alignment compared to our method. However, in experiment 3 (summary task), we observe that SFT struggles to align the distribution (refer to Figure 3), whereas our method successfully achieves alignment between the distribution of rewards for model-generated summaries and the distribution of rewards for human summaries.
>
> **3. Generalization of the method to large datasets**
> This paper aims to present a straightforward solution for training a model to generate outputs whose rewards align with a specified distribution. As discussed in the paper, our current focus is exclusively on distributions derived from data without addressing how to extrapolate the target. Your comment is very important as it highlights the most significant challenge of our method. In forthcoming versions of the paper, we will take care to elaborate on the process of extrapolating targets, either within a discussion or through additional experiments. One approach to broaden the scope across a larger dataset involves learning to map a prompt to a target. This can be achieved directly through a regression task or indirectly by assigning the target of a closely related prompt based on embedding approaches.
>
> **4. Name of the method**
> For an individual training prompt level, the target is a singular scalar. However, when considering the dataset as a whole, we obtain a distribution of rewards. Referring to this as the "target distribution method" serves as a means to distinguish the optimal distribution we target from the solution of the standard RL problem, which is the argmax over reward values.

---

> > ### Comment · Reviewer_xFiE · 2023-11-22
> >
> > Thank you for your responses. I apologize that I missed the comparison with SFT. I also have the follow-up questions regarding the responses:
> >
> > 1. Regarding the overoptimization phenomenon, I think Gao et al. has made the point that optimizing "too much" against the proxy reward might hurt the performance. However, if we do early stopping according to some criterion, we might also arrive at some point that aligns well with the ground truth. For example, I think one potential choice of "ground truth reward" for Figure 1 might be some sort of distance between the human sentence distribution and model output distribution. And it would be great if you can plot how the distance changes as we continue to optimize $R$ or $R_{reg}$. This might potentially reproduce the overoptimization phenomenon you want. Simply optimizing $R$ to the end does not help illustrate this point too much, since Gao et al. also found that the KL-reward frontier does not change too much even if we do not have KL regularization.
> >
> > 2. Could you please provide more discussions on why we want to match the human sentence distribution? It might happen that human responses are of mixed quality, and RLHF does help identifying the most high-quality ones, which may concentrate around high reward points. This is also related to the extrapolation question.
> >
> > 3. Could you please provide more details on the dataset used for SFT? Are you using all the preferred responses from the preference data, or using a different dataset?
> >
> > Overall, I think the paper studies a very important question and has made some good initial progress. But there is a bit more works to do before the paper could be finally published. So I keep the score toward reject for this round.

---

### Official Review · Reviewer_d8qz · 2023-11-05

**Soundness:** 2 fair
**Presentation:** 2 fair
**Contribution:** 2 fair
**Rating:** 3
**Confidence:** 4

**Summary:**

Reinforcement learning (RL) has emerged as an important tool for aligning large language models (LLM) with human preferences. Although RL has seen a lot of success, there are cons to using RL for alignment, which can have harmful side effects if not the alignment is not performed carefully. The most popular technique for mitigating this harmfulness is incorporating KL regularization into the objective. This paper proposes an additional technique for mitigating this harmfulness that empirically performs better than the KL regularization.

The central idea of the proposed technique is to collect a distribution of generations given a prompt and optimize the average reward of these generations given a prompt. The intuition is that designing a reward function on a collection of single prompt-generation pairs versus multiple generations per prompt prevents the reward function from being easily exploited. When an LLM exploits the reward, this leads to the harmful side effects.

**Strengths:**

Reinforcement learning from human preferences is a critical topic, given the impact of real-world applications. There are several strengths of the paper outlined below:


- Proposed solution: The idea of addressing issues with the reward by altering the reward function instead of regularizing the policy is a strength. In particular, the authors suggest collecting additional generation per prompt and using that as a regularization for the reward. Most attempts to address reward over-optimization either clip the reward score, add regularization to the reward training procedure, or add additional regularization to the policy optimization but have not focused on regularizing the reward  when optimized with RL


- Diversity of experiments: The authors perform experiments on three settings where issues can arise when optimizing LLM: calibrating sequence-level log-likelihood, over-specialization with a reward model, and multi-objective calibration. For each setting, the authors provide a thorough investigation of the issues of the baselines and reasons why their proposed approach is better.

**Weaknesses:**

Although the authors are addressing an important problem from an interesting perspective, the paper has several weaknesses outlined below:


- Clarity: This paper has several grammatical typos, undefined variables, and not well-explained concepts (see the question section). Furthermore, the use cases are repetitive; in particular, over-specialization and sequence-level likelihood captures the same issue. If the model sequence-level likelihood starts to diverge, that indicates the over-specialization problems, and vice versa. Furthermore, the multi-objective use case is not a traditional objective optimized in RLHF, and for a fixed alpha, the multi-objective can be viewed as a single objective. This means that this use case is similar to the previous use cases.


- Baselines: This paper is missing a few baselines, which makes it hard to understand the benefits of the proposed approach empirically( see the question section).

**Questions:**

Conceptual Questions:
1. What is the difference between reward overoptimization, reward hacking, and over-specialization?
2. What does it mean for a single prompt to have multiple generation pairs? Does this mean that the human is providing suboptimal generations?
3. What do you mean by target distribution? Why would a single prompt have a target distribution versus an optimal target?
4. What is the difference between training a policy to maximize the KL divergence between some initial policy \pi_0 and maximizing the sequence-level log-likelihood of \pi_0?


Technical Questions:
1. When you defined D := (x, t(x)) - what is t(x)?
2. How is the target distribution learned from single-prompt multiple-generation pairs? Is it the average of the generation rewards?
3. How do you build a prompt-continuation dataset based on public datasets? For example, in IMDB, you have a set of movie reviews, but merely grouping reviews using their first 10 tokens is a heuristic that is not generalizable. In particular, this observation would be an artifact of the prompt length; as you increase the prompt length, the overlap of generations would decrease.
4. For a given prompt x, the dataset consists of a set of generations for that prompt, i.e. {(x,y_i)}_{i=1}^{n}, where n is the number of generations per prompt. If this is the setting, then in equation 3, what is y? Is it a particular prompt in the set of prompts for a generation? If so, how do you choose which prompt to compare to the LLM-generated response \hat{y}?
5. Step 2 defines prompt data with a tuple instead of a set of tuples, meaning the dataset only has one prompt-generation pair instead of a set of prompt generations.
6. For minimizing the sequence-level log-likelihood, the most straightforward thing to do is to optimize the InstructGPT objective PPO-ptx [1], which includes a pretraining loss into the RL objective. How does your proposed algorithm compare to simply including that objective in the RL optimization procedure?
7. For the calibration experiments, I am not sure if the benefit is coming from the change in the reward function or the additional samples that you collect. It seems the reason why your algorithm is performing better is because you are taking the difference between y and \hay{y} - which provides a fixed target to optimize. But this situation does not generalize to the preference dataset when you don't have a ground truth y. Instead, you have comparison data; it is not obvious what the ground truth target would be. I am unsure how this use case is representative of the RLHF problems.
8. The results for calibration show various sampling schemes' performance, but what sampling scheme did you choose for your algorithm and the baseline RL algorithm?
9. What is the difference between your alignment loss and MLE?
10. How does R_reg perform on the multi-objective experiments?
11. Did you tune \alpha in the multi-objective experiments? If so how did you tune \alpha?
12. In Figure 4, the metrics arrows conflict with the best scores. For example, R^{pref} arrow it points upward, and the best score is 5.64, but you bolded 4.15. This issue is consistent with a few other metrics as well.

---

> ### Author Response · Authors · 2023-11-21
> **Part 1**
>
> We would like to thank the reviewer for their insightful comments. Those comments are valuable to improve further versions of the paper. We here respond to all comments.
>
> **Weaknesses**
>
> *Motivation*
>
> We apologize for any lack of clarity and appreciate the reviewer's feedback. We will carefully consider all of the provided comments to enhance the clarity of our work. Regarding the perceived repetitiveness of the experiments, we respectfully disagree. While it is acknowledged that these experiments revolve around the common theme of addressing over-optimization, each one is motivated by distinct reasons:
>
> - Experiment 1: The exploration of sequence-level log-likelihood serves as an illustration to emphasize the necessity of calibrating the RL objective. Drawing inspiration from Holtzman et al. (2019), who demonstrated the effectiveness of targeting a calibrated log-likelihood in the context of language (using temperature and top-p), our approach extends this idea to RL objective training. The sequence-level log-likelihood experiment provides intuition into the nature of the problem.
> - Experiment 2: This experiment delves into the standard scenario where a classifier reward model may undergo over-optimization. This problem is a common topic in the literature and highlights the relevance of our method in mitigating the impact of over-optimizing a proxy and generating unnatural outputs.
> - Experiment 3: We respectfully disagree with the reviewer's perspective on the significance of studying a multi-objective scenario. This is a common issue in RLHF, particularly in addressing the helpfulness/safety calibration challenge. In general, the challenge in RLHF is not to merely optimize a single objective with a fixed alpha but rather to determine the appropriate alpha. Our method addresses this challenge by defining targets for each objective based on data, offering a systematic approach rather than manually setting a fixed alpha and conducting a parameter search.
>
> *Baselines*
>
> We have endeavored to introduce a range of baselines, including diverse sampling procedures for Experiment 1 and comparisons with standard optimization algorithms for Experiments 2 and 3 (such as supervised fine-tuning, standard reinforcement learning, and reinforcement learning with KL divergence). Nevertheless, we acknowledge that this may be perceived as insufficient, and we are eager to incorporate additional baselines. What recommendations do you have in this regard?
>
> **Conceptual questions**
>
> 1. We regret any imprecision in our use of terminology. Our primary intention is to align with the definition of *over-optimization* as articulated by Gao et al. 2023:
>
> "In machine learning, this effect arises with proxy objectives provided by static learned models, such as discriminators and reward models. Optimizing too much against such a model eventually hinders the true objective, a phenomenon we refer to as over-optimization."
>
> In subsequent iterations of the paper, we will exercise greater care in the precise application and definition of these terms.
>
> 2. We unfortunately believe there might be a misunderstanding regarding our methodology that hinder the overall understanding of the paper. We never consider a single prompt generating multiple pairs of generations. In Section 3.2, when outlining our methodology, we specifically refer to pairs as "prompt-continuation," "prompt-reward," or "target reward." We acknowledge that this misunderstanding may stem from a lack of clarity in our method and notations. We are committed to enhancing the clarity of this section to avoid any confusion that may have led to a misunderstanding of the paper. We sincerely apologize for any confusion that may have occurred.
>
> 3. For a single prompt, we define a single target. When considering these targets across the entire dataset, a distribution of rewards emerges. We employ the term "distribution" at the dataset level to contrast with the standard RL optimal solution, which is the argmax distribution over the reward values. We will clarify this notion in future iterations of the paper.
>
> 4. It can indeed be very similar if the KL is computed at the sentence level.

---

> > ### Author Response · Authors · 2023-11-21
> > **Part 2**
> >
> > **Technical**
> >
> > *The first questions are related to the aforementioned mis-understanding.*
> >
> > 1. As defined in section 3.2, t(x) is the reward obtained by the human “ground truth” generation R(x,y). To enhance clarity, we will explicitly state: t(x)=R(x,y), where where x is the prompt and y the human continuation.
> > 2. Each prompt corresponds to a single generation. The model is trained to align with a target distribution at the dataset level.
> > 3. With the previous clarifications, this question does not hold in the context of a single prompt-target.
> > 4. We exclusively refer to a single generation y per prompt x and never mention several y_i.
> > 5. Yes, this is the case for the entire paper.
> >
> > *Other questions*
> >
> > 6. All our RL objectives are optimized using PPO, so optimizing R with PPO corresponds to the objective you mention. Results for this objective are presented in Figure 1b (top right).
> >
> > 7. Your comment is insightful and is related to one of the main challenges of this target approach. In this paper, we do not collect additional data but construct targets based on the additional confidence annotations from RLHF annotators. Therefore, our training is focused on a subset of available comparisons, which proved sufficient for obtaining valuable results in the studied case. To extend this approach to a larger dataset, we would suggest extrapolating the target by learning to map a prompt to a target (either directly through a regression task or indirectly by assigning the target of a similar prompt based on embedding approaches). In future versions of the paper, we plan to include those remarks on target extrapolation into the discussion or even build an additional experiment.
> >
> > 8. For all our experiments, the decoding algorithm involves sampling with a temperature and top-p set to 1. In the case of sequence-level likelihood, training our objective allows to target the human sequence level log-likelihood without having to adjust the sampling procedure: this sampling procedure is automatically learnt.
> >
> > 9. Instead of doing MLE, we target the setence level log-likelihood of human sentences.
> >
> > 10. For the single objective, we tuned the parameter and present the results in the table. Regarding the multi-objective, we chose not to display the results as it posed challenges in optimization, involving a 3-objective optimization (optimizing the two rewards + regularization). We also decided against displaying the results, as the two rewards effectively regulate each other, limiting direct overfitting and the need of adding a KL regularization.
> >
> > 11. We tuned alpha by conducting an overall search to maximize the average alignment of the two rewards. We will clarify this methodology in the future versions of the paper.
> >
> > 12. The highlighted numbers are indeed a typo spread throughout the paper. The intention was to highlight methods that reach values close to the human baseline. We apologize for these errors and will ensure to correct them in future versions of the paper.

---

### Meta-Review · Area_Chair_vW7P · 2023-12-19

**Metareview:**

The paper presents a novel approach to mitigating harmful side effects in reinforcement learning for aligning large language models with human preferences. Reviewers appreciated an interesting idea of modifying the reward function and the diversity of experiments. However, concerns were raised about significant clarity issues, lack of comprehensive baselines, and methodological issues. Due to these reasons, I do not recommend acceptance at this time. I encourage the authors to focus on improving clarity, adding more comprehensive baselines, and addressing the methodological concerns to strengthen the paper.

**Justification For Why Not Higher Score:**

Clarity issues, lack of comprehensive baselines.

**Justification For Why Not Lower Score:**

N/A

---

### Decision · Program_Chairs · 2024-01-16

Reject